# A Study on Visually Induced VR Reduction Method for Virtual Reality Sickness

Ju-hye Won [ID] and Yoon Sang Kim *[ID]

BioComputing Lab, Department of Computer Science and Engineering, Institute for Bio-Engineering Application Technology, Korea University of Technology and Education (KOREATECH), Cheonan 31253, Korea; wonjh1206@koreatech.ac.kr
* Correspondence: yoonsang@koreatech.ac.kr

**Abstract:** In this paper, we propose a new type of visually induced reduction method for virtual reality (VR) sickness. This method induces a gaze based on a visual guide (VG). Although VGs are used in commercial VR game contents as crosshairs, no studies have been conducted related to VR sickness. VGs can have various properties determined by position, size, shape, color, etc., and it was necessary to investigate which properties affect VR sickness. In particular, size and position of VG are properties that directly affect the user's gaze movement. Therefore, in this paper, five VR sickness reduction methods with different position and size of VG are proposed. Then, an experiment was conducted to investigate the effectiveness of the proposed VR sickness reduction method. To this end, a SSQ (including nausea, oculomotor discomfort, disorientation, and total score) and a questionnaire on fatigue and immersion were used. From the experimental results, a VG with a size 30% that of the aspect ratio, and a position synchronized to the user's head movement direction, was most effective in terms of VR sickness reduction and immersion.

**Keywords:** VR sickness; VR sickness reduction method; SSQ; visual guide; crosshair; gender





## 1. Introduction

Recently, the virtual reality (VR) industry has developed rapidly due to the spread of various head-mounted display (HMD) devices. However, VR sickness that occurs when using VR HMDs is a major concern which could interrupt the spread of VR content. There have been various suggested causes of VR sickness, but they have not been clearly identified yet. Therefore, much research has been conducted to identify the cause of VR sickness and to reduce it in terms of device, content, and human factors. In particular, studies have been conducted to reduce VR sickness in terms of general-purpose content that is not dependent on VR HMDs.

Researchers at HBRS Sankt Augustin (Germany) conducted a study to reduce VR sickness by using the peripheral visual effects of VR videos [1]. Wienrich et al. reduced VR sickness by applying a virtual human nose as an earth-fixed grid to the content [2]. Studies have also been conducted to evaluate content quality and VR sickness symptoms in two HMD environments, Oculus Rift and HTC Vive, and compare VR sickness symptom differences according to resolution; the researchers concluded that HTC Vive provides better integrated quality for VR sickness reduction [3]. Meanwhile, Nie et al. conducted VR sickness reduction studies using users' gaze induction based on real-time video processing [4]. Also, researchers at Columbia University (New York, NY, USA) have reduced VR sickness with dynamic FOV modification, which partially limits the user's field of view (FOV) [5]. Another study was conducted to reduce discomfort in a way that regulates the depth of VR videos in real-time [6]. Whittinghill et al. conducted a study to reduce VR sickness by showing a visual reference point on VR roller coasters [7]. Furthermore, researchers at University of Nottingham (Nottingham, UK) found that the higher cognitive level at which users interact with VR, the less resulting VR sickness, showing that there is

a correlation between VR interaction and VR sickness symptoms according to cognitive level [8]. Also, 'Virtual Guiding Avatar', that combines various motion properties with an independent visual background (IVB), was used to reduce VR sickness [9].

Existing studies have mainly dealt with VR sickness caused by camera speed, complexity of background, and unpredictable object movement in VR HMD environments. Studies have also tried to reduce VR sickness by artificially inducing users' gaze through FOV and IVB. However, these methods have a problem in that they reduce VR fidelity by causing a sensory conflict between content and users. To solve this, it is necessary to provide a new visually induced reduction method that can reduce VR sickness while maintaining VR fidelity. Therefore, in this paper, we propose a new type of visually induced VR sickness reduction method using a visual guide (VG) that can induce a user's gaze. Although VGs are used in commercial VR game contents as crosshairs, there have been no studies conducted related to VR sickness. This VG can have various properties determined by position, size, shape, color, etc., and it is necessary to investigate which properties affect VR sickness.

Thus, in this paper, VR sickness reduction methods based on VGs with different properties are designed, and an experiment is conducted to verify the effectiveness of VR sickness reduction methods. The experimental results are analyzed using a simulator sickness questionnaire (SSQ: nausea, oculomotor discomfort, disorientation, and total score) [10,11] and a questionnaire on fatigue and immersion.

## 2. Materials and Methods

### 2.1. Design of a Visually Induced VR Sickness Reduction Method

This section describes a visually induced VR sickness reduction method based on a VG. A VG is an independent element in the form of a guide to visually induce gaze movement. This has a role in reducing the level of a user's concentration on VR sickness' causes: fast movement and rotation speed of the camera, the complexity of the background, unpredictable object movement, and fast screen transitions. This VG can have various properties in position, size, shape, color, etc. However, it is difficult to determine which of these various properties are most effective for VR sickness reduction In commercial VR games, a VG of white color, circle shape, and 1 pixel line thickness was commonly used. Also, the main properties that directly affect the user's gaze movement, position and size, were being used in a wide variety of ways. Thus, the color, shape, and the thickness of VG to be used for VR sickness reduction methods were fixed to white, circle, and 1 pixel based on existing references. The remaining two properties (size and position) were set as parameters of VR sickness reduction method.

We confirmed from a pilot study [12] that among the various sizes of VG, a size 30% that of the aspect ratio [13] was more effective in reducing VR sickness than other sizes. Therefore, based on this result, five VR sickness reduction methods were designed by changing the two properties of VGs which have a size 30% that of the aspect ratio. First, a VG was designed to maintain size 30% that of the aspect ratio or designed to vary its size from 0% to 30% that of the aspect ratio. Then, a VG was designed to have a position synchronized with the user's head or gaze movement direction or moving along in the clockwise (CW)/counterclockwise (CCW) direction.

The VG of the first VR sickness reduction method had a size 30% that of the aspect ratio, and a position synchronized to the user's head movement direction. Figure 1 shows the first VR sickness reduction method (M1).

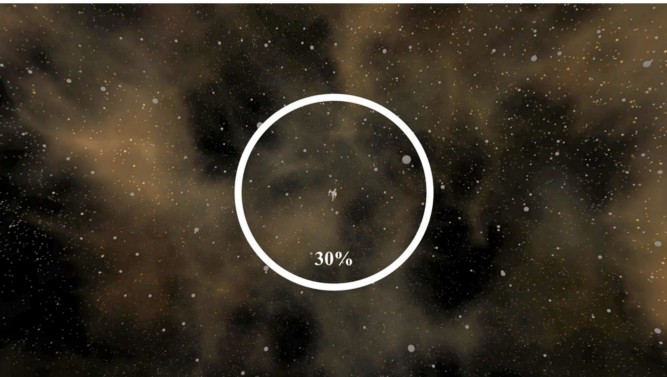

**Figure 1.** The first VR sickness reduction method (M1).

The VG of the second VR sickness reduction method had a size that changes from 0% to 30% that of the aspect ratio, and a position synchronized to the user's head movement direction. This was a way to temporarily attract the user's gaze. In other words, as the VG was no longer visible when the size of the VG reached 0%, this allowed users to focus more on the content. Figure 2 shows the second VR sickness reduction method (M2).

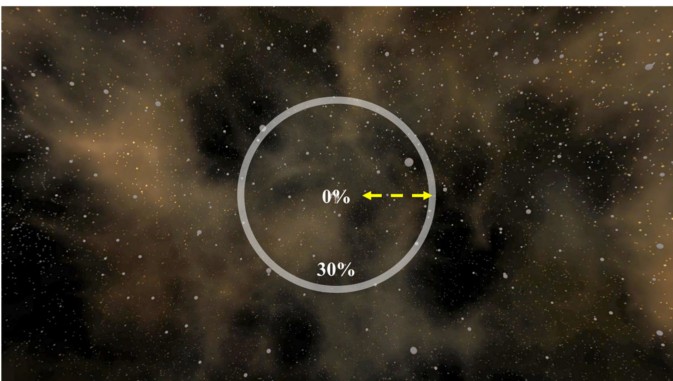

**Figure 2.** The second VR sickness reduction method (M2).

The VG of the third VR sickness reduction method had a size 30% that of the aspect ratio, and a position that moves CW every second along the eight points. This method induced CW gaze movement through the movement of the VG. Figure 3 shows the third VR sickness reduction method (M3).

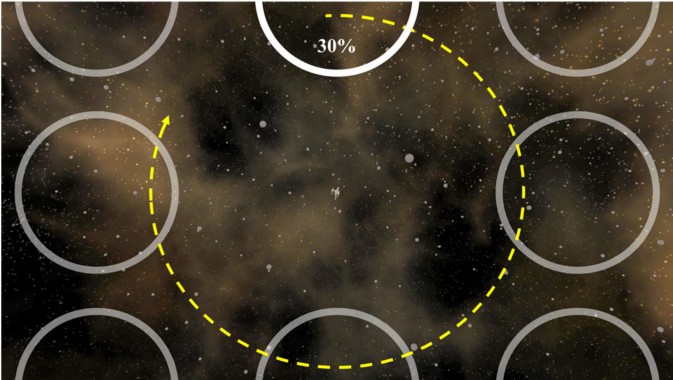

**Figure 3.** The third VR sickness reduction method (M3).

The VG of the fourth VR sickness reduction method had a size 30% that of the aspect ratio, and a position that moves CCW every second along the eight points. This method

induced CCW gaze movement through the movement of the VG. Figure 4 shows the fourth VR sickness reduction method (M4).

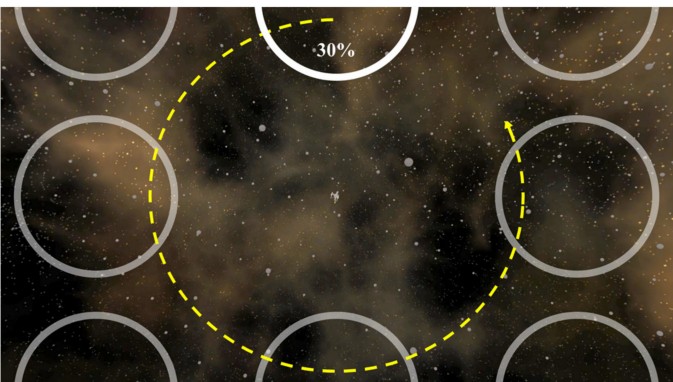

**Figure 4.** The fourth VR sickness reduction method (M4).

The VG of the fifth VR sickness reduction method had a size 30% that of the aspect ratio, and a position synchronized to the user's gaze movement direction. This method continuously induced the user's gaze by positioning the VG in the user's gaze direction. Figure 5 shows the fifth VR sickness reduction method (M5).

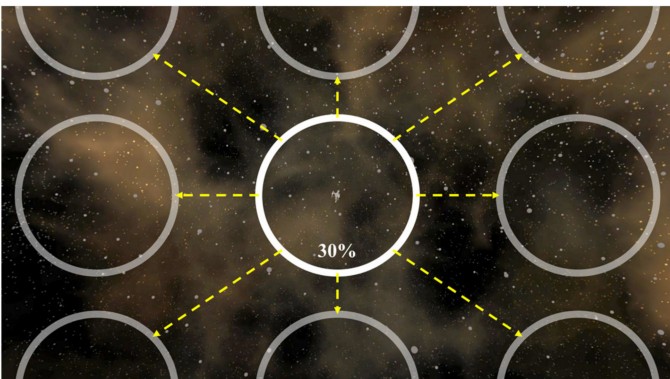

**Figure 5.** The fifth VR sickness reduction method (M5).

### 2.2. Experimental Environment

This section describes the experimental environment used to verify the effectiveness of the designed VR sickness reduction methods. The experiment was conducted on 40 subjects in their twenties and thirties (male: 20, female: 20). In the experiment, a 3D VR space-flight video that caused VR sickness was used. This video used for the test was self-produced to cause VR sickness using the camera's acceleration, deceleration, and rotation (yaw, pitch, roll). Considering that the level of VR sickness increases when exposed to VR content for a long time, the play time of VR video was set to 60 s [14]. Using HTC Vive Pro Eye [15], subjects watched videos with each of the VR sickness reduction methods described in Section 2.1. Figure 6 shows the experimental environment. Table 1 shows the VR sickness reduction method used in experiment.

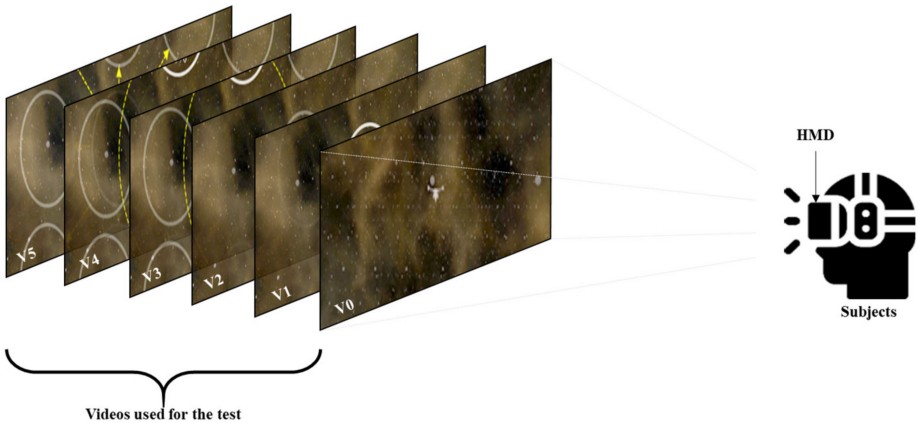

**Figure 6.** Experimental environment.

**Table 1.** VR sickness reduction method used in the experiment.

| Method | Properties |
|--------|-----------|
| M0 | Without VG |
| M1 | Size: 30% of aspect ratio<br>Position: Movement with head tracking |
| M2 | Size: 0%↔30% of aspect ratio (every second)<br>Position: Movement with head tracking |
| M3 | Size: 30% of aspect ratio<br>Position: Movement with CW (every second) |
| M4 | Size: 30% of aspect ratio<br>Position: Movement with CCW (every second) |
| M5 | Size: 30% of aspect ratio<br>Position: Movement with eye tracking (every second) |

Before the experiment, subjects answered the SSQ for sickness condition measuring and performed calibration for eye-tracking after wearing VR HMD. In the experiment, a total of six videos (V0: original video; V1–V5: videos with VR sickness reduction methods using VGs) were used. At the end of each video, which was provided for 60 s, subjects had time to respond to the SSQ and the questionnaire on fatigue and immersion. The questionnaire on fatigue and immersion was conducted by subjects answering whether they felt fatigue or immersion for V1–V5. After answering the questionnaire, subjects had a rest time to relax VR sickness. Also, the original video (V0) without VR sickness reduction method and videos with VR sickness reduction method (V1–V5) were randomly placed to ensure reliability. Experiments were conducted using the protocol shown in Figure 7 for each VR HMD video. Also, in Figure 7, the V(x) symbol was used to represent the randomly placed videos (V0–V5).

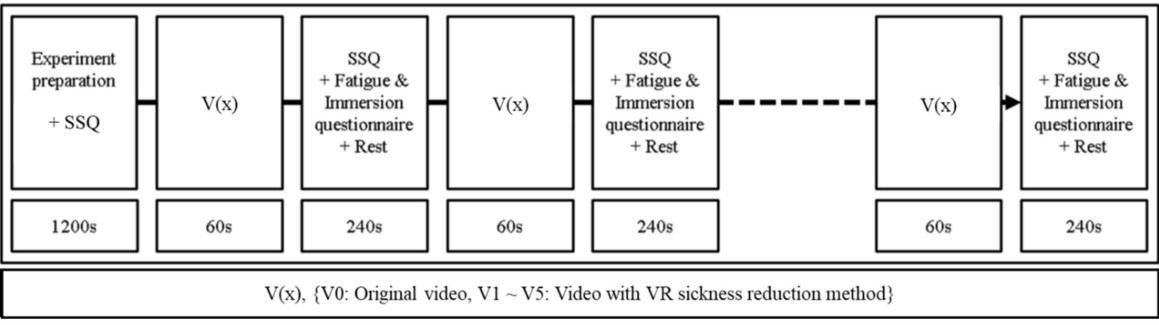

**Figure 7.** Experimental protocol.

## 3. Results

In this section, we analyze the experimental results to confirm the effectiveness of the proposed VR sickness reduction methods and discuss. First, using the Shapiro–Wilk test, we confirmed that all measured SSQ data satisfy the normal distribution. Then, SSQ measured in the original video (V0) without VR sickness reduction method and SSQ measured in videos with VR sickness reduction method used (V1–V5) were compared by the paired *t*-test. Nausea score was significant when the M1 was used ($p = 0.042$): nausea was reduced by 36%. When the other VR sickness reduction methods (M2–M5) were used, nausea was not significantly reduced. From this result, it was confirmed that a continuously expressed VG without movement significantly reduces nausea. Oculomotor discomfort score was not significant in any VR sickness reduction methods (M1–M5). Since the proposed methods (M1–M5) were a method to reduce VR sickness by inducing the user's gaze, it could not confirm the effect of reducing oculomotor discomfort. Disorientation score was significant when the M1 ($p = 0.019$), M2 ($p = 0.002$), and M4 ($p = 0.048$) were used: disorientation was reduced by 33%, 43%, and 30% in M1, M2, and M4, respectively. From these results, most proposed methods were confirmed to reduce disorientation. Total score was significant when M1 was used: total score was reduced by 29%. From these results, it was confirmed that the M1 was only method to reduce VR sickness level overall. Table 2 shows a comparison of VR sickness reduction effects.

**Table 2.** VR sickness reduction method used in experiment.

| Method | SSQ | | | | | | | |
|---|---|---|---|---|---|---|---|---|
| | Nausea (N) | | Oculomotor Discomfort (O) | | Disorientation (D) | | Total Score (T) | |
| | Score | Rate (*p*-Value) | Score | Rate (*p*-Value) | Score | Rate (*p*-Value) | Score | Rate (*p*-Value) |
| M0 | 13.36 | - | 21.41 | - | 26.45 | - | 22.91 | - |
| M1 | 8.59 | −36% (0.042 *) | 16.30 | −24% (0.129) | 17.75 | −33% (0.019 *) | 16.18 | −29% (0.044 *) |
| M2 | 11.93 | −11% (0.694) | 19.33 | −10% (0.613) | 14.96 | −43% (0.002 **) | 18.23 | −20% (0.233) |
| M3 | 10.49 | −21% (0.166) | 17.43 | −19% (0.092) | 29.93 | +13% (0.759) | 20.76 | −9% (0.598) |
| M4 | 13.12 | −2% (0.924) | 19.71 | −8% (0.560) | 18.44 | −30% (0.048 *) | 19.82 | −13% (0.338) |
| M5 | 12.40 | −7% (0.775) | 19.52 | −9% (0.606) | 18.79 | −29% (0.115) | 19.54 | −15% (0.425) |

\* $p < 0.05$. \*\* $p < 0.01$.

Figure 8 shows a graph comparing SSQ score of VR sickness reduction methods. M1 showed the greatest reduction in nausea, disorientation, and total score (Circle in Figure 8). Furthermore, M2 and M4 showed significant reduction in disorientation (Circle in Figure 8). Other VR sickness reduction methods such as M3 and M5 did not show significant reduction effect. Therefore, it was confirmed that the most effective method for reducing VR sickness is M1.

Additionally, we analyzed SSQ results of M0 and M1 using paired *t*-test to confirm the difference in the level of VR sickness depending on gender (M0: no VG; M1: most effective method). The analysis results show that the difference between the genders was not significant. Table 3 shows the difference in the level of VR sickness depending on gender.

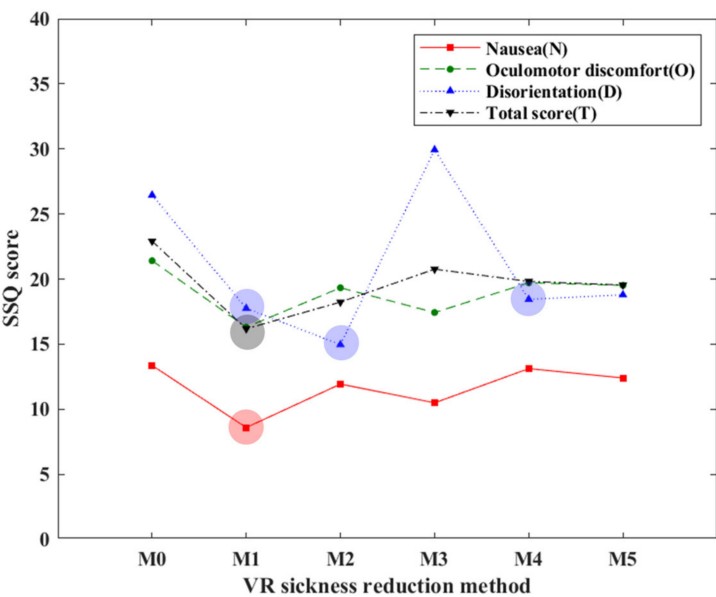

**Figure 8.** Comparing SSQ score of VR sickness reduction methods.

**Table 3.** The difference in the level of VR sickness depending on gender.

| Method | | Nausea (N) | | Oculomotor Discomfort (O) | | Disorientation (D) | | Total Score (T) | |
|---|---|---|---|---|---|---|---|---|---|
| | | Score | *p*-Value | Score | *p*-Value | Score | *p*-Value | Score | *p*-Value |
| M0 | Man | 14.78 | - | 23.49 | - | 27.14 | - | 24.68 | - |
| | Woman | 11.92 | 0.627 | 19.32 | 0.577 | 25.75 | 0.900 | 21.13 | 0.677 |
| M1 | Man | 7.63 | - | 15.16 | - | 14.61 | - | 14.39 | - |
| | Woman | 9.54 | 0.654 | 17.43 | 0.690 | 20.88 | 0.541 | 17.95 | 0.605 |

$p = 0.05$.

From the results of the fatigue questionnaire response, M4 was shown to be the most fatigue causing method. From the results, M4 was confirmed to be effective in reducing disorientation, but it was also confirmed that this method caused the most fatigue in subjects. In addition, M3 was confirmed to have the second highest fatigue score. This indicates that VR reduction methods that induce artificial gaze movement along CW/CCW increase fatigue. The results of the immersion questionnaire response show that the immersion level was the highest in M1. Therefore, it was confirmed that a continuously expressed VG without movement increased users' immersion. As a result, a VG with a size 30% that of the aspect ratio, and a position synchronized to the user's head movement direction, was most effective in terms of VR sickness reduction and immersion. Table 4 shows the questionnaire results for fatigue and immersion. The higher the value, the more subjects felt fatigue or immersion for each method.

**Table 4.** The questionnaire results for fatigue and immersion.

| Method | Fatigue | Immersion |
|---|---|---|
| M1 | 0 | 13 |
| M2 | 1 | 4 |
| M3 | 7 | 3 |
| M4 | 8 | 3 |
| M5 | 5 | 4 |

Figure 9 shows a graph comparing fatigue and immersion scores of VR sickness reduction methods. While M4 had the lowest immersion and the highest fatigue (Red circle

in Figure 9), M1 had the highest immersion and the lowest fatigue (Black circle in Figure 9). These results indicate that M1 has an effect on VR sickness reduction and induce to relatively low fatigue and high immersion.

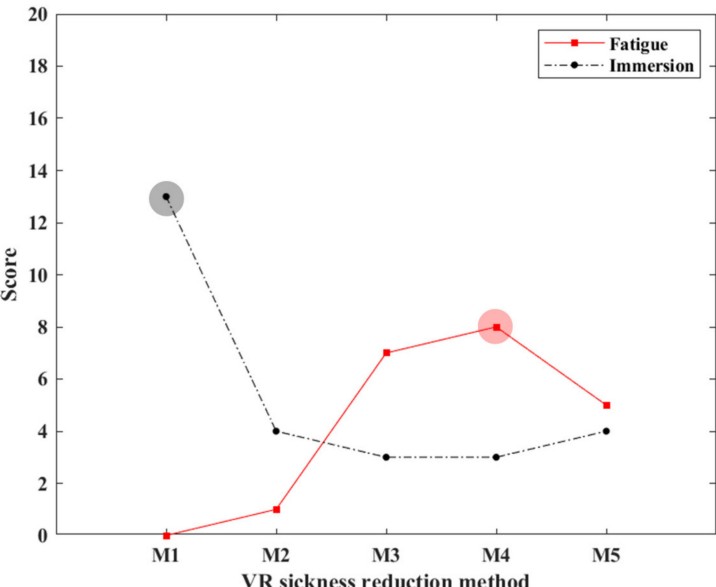

**Figure 9.** Comparing fatigue and immersion scores of VR sickness reduction methods.

## 4. Conclusions

In this paper, we proposed a new type of visually induced VR sickness reduction method. The proposed method was to reduce VR sickness by inducing the user's gaze based on a VG. In this paper, five VR sickness reduction methods with different size and position of VG were designed, and experiments were conducted to confirm the effectiveness of these methods. To compare the effects of VR sickness reduction methods, SSQ scores and a questionnaire on fatigue and immersion scores were used. From experimental results, VR sickness was significantly reduced when a VG (M1) with a size 30% that of the aspect ratio, and a position synchronized to the user's head movement direction, was used. In addition, users were confirmed to show the highest immersion and lowest fatigue. Therefore, we confirmed that a new type of visually induced VR sickness reduction method could be the method designed using a VG which includes the properties of M1. It is expected that our proposed method could work for reducing VR sickness and contribute to the spreading of VR content.

**Author Contributions:** Conceptualization, Y.S.K.; methodology, Y.S.K.; software, J.-h.W.; validation, Y.S.K.; formal analysis, J.-h.W.; investigation, Y.S.K.; resources, Y.S.K.; data curation, J.-h.W.; writing—original draft preparation, Y.S.K.; writing—review and editing, Y.S.K.; visualization, J.-h.W.; supervision, Y.S.K.; project administration, Y.S.K.; funding acquisition, Y.S.K. All authors have read and agreed to the published version of the manuscript.

**Funding:** This work was supported by the National Research Foundation of Korea (NRF) grant funded by the Korea government (MSIT) (No. NRF-2020R1F1A1076114).

**Institutional Review Board Statement:** This study was conducted according to the guidelines of the Declaration of Helsinki and approved by the Institutional Review Board of KOREATECH (approved on 4 September 2019).

**Informed Consent Statement:** Informed consent was obtained from all subjects involved in the study.

**Data Availability Statement:** Not applicable.

**Conflicts of Interest:** The authors declare no conflict of interest.

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
