# Peer review of "A Study on Visually Induced VR Reduction Method for Virtual Reality Sickness"

_applsci, doi:10.3390/app11146339_

Round 1

Reviewer 1 Report

The article presents a timely and an interesting research.
Although, as I am not only a researcher, but one who plays video games quite often, I believe these results "could be expected". But, as written in the article, no-one covered this subject of crosshairs in virtual reality (I could not find similar studies at the moment). Therefore, it can be counted as new scientific research.

The authors mentioned in the Materials and Methods section that "We confirmed from a pilot study that among the various sizes of VG, a 30% size of the aspect ratio [14] was more effective in reducing VR sickness than other sizes.".

I looked into this study, but I could not find anything about 30% size of the aspect ratio. It only talks about the field of view, and an increase in immersion between approximately 60-90° of FoV. While the difference is about 30°, I do not think that these two parameters and values are the same.

Results:
The authors used the paired t-test to search for significant differences, however before using such tests, the distribution of data should be investigated (Kolmogorov-Smirnov test / Shapiro-Wilk test / visual methods such as QQ-plots, etc.). The authors should be careful, because if the data distribution is not normal, non-parametric tests should be used (the t-test is parametric).

Also, other studies have shown that (depending of the tasks, of course) women react to / experience VR differently, than men. Some of their senses are more hightened. Since results of women can be found in your data, you could search for differences between the results of men and women. It is possible that one of the two tested genders had a significantly different VR experience than the other. It would be interesting to see whether this difference exists, and it would greatly increase the scientific value of this study.

English:
The text requires moderate changes. While the sentences are understandable, sometimes definite and indefinite articles are missing from them, and a few of them do not have subjects.
From line 32 to 51, every sentence starts with the word "Researchers" (9 sentences, if I counted correctly). Almost every sentence in this interval is structured as the following: researchers from X studied Y (to achieve Z). While it is not exactly a problem, it feels redundant and repetitive. The word "researchers" is also repeated in some sentences. Other words are also repeated in this article, see if you can find some synonyms for them.
There are a few typos as well (e.g. Oculus "lift").

References:
The references are not in the style of the MDPI journals and are in an inconsistent format.

Author Response

Responses to Reviewer:

First of all, the authors would like to express sincere gratitude for reviewer’s invaluable time and thoughtful comments.

In order to save the reviewer's invaluable time, and expedite the processing of the revised manuscript, the authors tried to be as specific as possible in our response to the reviewer.

Overall, the authors agreed with the reviewer's opinion and comment. For better quality of the paper, the authors had read the manuscript carefully, and some sentences, figures were added and modified according to the reviewer's comment.

Response to Reviewer 1 Comments

Points 1. The authors mentioned in the Materials and Methods section that "We confirmed from a pilot study that among the various sizes of VG, a 30% size of the aspect ratio [14] was more effective in reducing VR sickness than other sizes.".

I looked into this study, 1)but I could not find anything about 30% size of the aspect ratio. It only talks about the field of view, and an increase in immersion between approximately 60-90° of FoV. While the difference is about 30°, 2)I do not think that these two parameters and values are the same.

(Answers)

Thanks for the invaluable comment.

The reference [14] was added to explain the word ‘aspect ratio’. Therefore, readers could not find results of our pilot study (a 30% size of the aspect ratio was more effective in reducing VR sickness.). To make clear this context, we have added our pilot study paper (to be published) as a reference (we did not add it as a reference at first submission).

Please refer the blue texts: Page 2 line 82 and Page 9 line 270

Points 2. The authors used the paired t-test to search for significant differences, however before using such tests, the distribution of data should be investigated (Kolmogorov-Smirnov test / Shapiro-Wilk test / visual methods such as QQ-plots, etc.). The authors should be careful, because if the data distribution is not normal, non-parametric tests should be used (the t-test is parametric).

(Answers)

As the reviewer noticed us, we found that experimental data required an additional analysis. We appreciate for these invaluable comments. As the reviewer suggested, we have conducted the Shapiro-Wilk test, and confirmed that the data satisfies the normal distribution. Thus, we used paired t-test for data analysis. Also we have added this result in the ‘Result’ section. Please refer the blue texts: Page 6 line 155-156

Points 3. Also, other studies have shown that (depending of the tasks, of course) women react to / experience VR differently, than men. Some of their senses are more hightened. Since results of women can be found in your data, you could search for differences between the results of men and women. It is possible that one of the two tested genders had a significantly different VR experience than the other. It would be interesting to see whether this difference exists, and it would greatly increase the scientific value of this study.

(Answers)

Thanks for this invaluable comment! According to reviewer’s suggestion, we have conducted an additional analysis to examining the difference in the level of VR sickness between men and women, and found a significant result as the below:

Additionally, we analyzed SSQ results of M0 and M1 using paired t-test to confirm the difference in the level of VR sickness depending on gender (M0: no VG, M1: most effectiveness method). The analysis results showed that the difference between the genders was not significant.

We have added this new result in the ‘Result’ section. Please refer the blue texts: Page 7 line 183-188

Points 4. The text requires moderate changes. While the sentences are understandable, sometimes definite and indefinite articles are missing from them, and a few of them do not have subjects.

From line 32 to 51, every sentence starts with the word "Researchers" (9 sentences, if I counted correctly). Almost every sentence in this interval is structured as the following: researchers from X studied Y (to achieve Z). While it is not exactly a problem, it feels redundant and repetitive. The word "researchers" is also repeated in some sentences. Other words are also repeated in this article, see if you can find some synonyms for them.

There are a few typos as well (e.g. Oculus "lift").

(Answers)

Thanks for the considerable comment. According to reviewer’s comment, we modified the part that explains the previous researches in ‘Introduction’ section by referring to the introduction progression of papers in Applied Sciences Journal. Please refer the blue texts: Page 1 line 32-49

And we have modified overall sentences for understandable.

Points 5. The references are not in the style of the MDPI journals and are in an inconsistent format.

(Answers)

Thanks for the considerable comment. According to reviewer’s comment, we have modified the references following the style of the MDPI journals. Please refer the blue texts: Page 9 line 242-275

Reviewer 2 Report

The paper deals with a very important issue with VR - the so called Virtual Reality Sickness, which hasn't been solved successfully so far. In this sense the paper represents further advancement in this area and I believe that the authors not only did a great research, but the results of which could inspire other teams to follow the and carry on the responsible work.

Author Response

Responses to Reviewer:

First of all, the authors would like to express sincere gratitude for reviewer’s invaluable time and thoughtful comments.

Reviewer 3 Report

The paper compared five methods to reduce visually-induced virtual reality sickness. 

Major comment:

1.  It appears that M1 is effective because it is without movement. It is common sense that continuous head movement induces nausea. Does this significantly reduce the scientific contribution of this research work? It simply proved something that is very much common sense. 

Author Response

Responses to Reviewer:

First of all, the authors would like to express sincere gratitude for reviewer’s invaluable time and thoughtful comments.

In order to save the reviewer's invaluable time, and expedite the processing of the revised manuscript, the authors tried to be as specific as possible in our response to the reviewer.

Overall, the authors agreed with the reviewer's opinion and comment. For better quality of the paper, the authors had read the manuscript carefully, and some sentences, figures were added and modified according to the reviewer's comment.

Response to Reviewer 3 Comments

Points 1. It appears that M1 is effective because it is without movement. It is common sense that continuous head movement induces nausea. Does this significantly reduce the scientific contribution of this research work? It simply proved something that is very much common sense.

(Answers)

Thanks for the considerable comment. Continuous head movement did not occur during the experiments, because the methods, including M1, did not induce user’s head movement. Therefore, the authors think the result of this study is significant.

Round 2

Reviewer 1 Report

Thank you for the correction.

I propose acceptation.